# Multi-Scene Mask Detection Based on Multi-Scale Residual and Complementary Attention Mechanism

**DOI:** 10.3390/s23218851

**Published:** 2023-10-31

**Authors:** Yuting Zhou, Xin Lin, Shi Luo, Sixian Ding, Luyang Xiao, Chao Ren

**Affiliations:** College of Electronics and Information Engineering, Sichuan University, Chengdu 610065, China; zhouyutingx@stu.scu.edu.cn (Y.Z.); linxin@stu.scu.edu.cn (X.L.); luoshi1@stu.scu.edu.cn (S.L.); dingsixian@stu.scu.edu.cn (S.D.); xiaoluyang@stu.scu.edu.cn (L.X.)

**Keywords:** multi-scene mask detection, deep learning, multi-scale residual, channel-spatial attention, masked face dataset, generalization improvement strategy

## Abstract

Vast amounts of monitoring data can be obtained through various optical sensors, and mask detection based on deep learning integrates neural science into a variety of applications in everyday life. However, mask detection poses technical challenges such as small targets, complex scenes, and occlusions, which necessitate high accuracy and robustness in multi-scene target detection networks. Considering that multi-scale features can increase the receptive field and attention mechanism can improve the detection effect of small targets, we propose the YOLO-MSM network based on the multi-scale residual (MSR) block, multi-scale residual cascaded channel-spatial attention (MSR-CCSA) block, enhanced residual CCSA (ER-CCSA) block, and enhanced residual PCSA (ER-PCSA) block. Considering the performance and parameters, we use YOLOv5 as the baseline network. Firstly, for the MSR block, we construct hierarchical residual connections in the residual blocks to extract multi-scale features and obtain finer features. Secondly, to realize the joint attention function of channel and space, both the CCSA block and PCSA block are adopted. In addition, we construct a new dataset named Multi-Scene-Mask, which contains various scenes, crowd densities, and mask types. Experiments on the dataset show that YOLO-MSM achieves an average precision of 97.51%, showing better performance than other detection networks. Compared with the baseline network, the mAP value of YOLO-MSM is increased by 3.46%. Moreover, we propose a module generalization improvement strategy (GIS) by training YOLO-MSM on the dataset augmented with white Gaussian addition noise to improve the generalization ability of the network. The test results verify that GIS can greatly improve the generalization of the network and YOLO-MSM has stronger generalization ability than the baseline.

## 1. Introduction

Deep learning for mask detection has important demand in medical and industrial production, which reflects the application of neural networks and image sensors in daily life [1,2]. During an epidemic of respiratory viruses, mask detection can effectively supervise the wearing of masks, thereby reducing the risk of virus transmission. In hospitals, doctors and patients are often required to wear masks under certain circumstances for their safety. Also, mask detection technology used in industrial production can prevent dust from entering the respiratory tract of workers. In December 2019, the novel coronavirus (COVID-19) swept the world, causing irretrievable casualties and enormous economic losses around the world. Therefore, during the epidemic period, basic protective measures against infection in public places need to be taken. The main mode of transmission of coronavirus is droplets and close contact [3], so wearing a mask plays a very effective role in slowing down the spread of the virus and reducing the risk of infection [4]. On the one hand, wearing a mask can control the spread of the virus from the source, which can effectively reduce the risk of droplets being transmitted by the infected person to the surrounding air [5]; on the other hand, masks can effectively protect uninfected people by filtering virus-containing particles in the air [6]. The importance of masks is not only reflected in protecting against respiratory virus infection, but also in areas such as industrial production.

For example, wearing masks can prevent workers from inhaling harmful gases and particles produced during production [7]. It can be seen that it is very necessary to establish an effective mask supervision and management mechanism. At present, the supervision of masks in public places is mostly manual, which is labor-intensive, and the effect is not satisfying. Therefore, the research on algorithms to automatically detect mask wearing is of great practical significance for the health and safety of citizens.

At present, deep learning is developing rapidly in the field of computer vision. Compared with object detection based on traditional machine learning, such as the V-J detection network [8], DPM network [9], HOG detection network [10], etc., object detection based on deep learning has significant advantages in object recognition ability and algorithm adaptability [11]. In 2021, Batagelj and Peer et al. constructed a large face image dataset called the Face-Mask-Label Dataset (FMLD) from the publicly available MAFA [12] and Wider Face datasets [13], and proposed a pipeline to recognize whether a mask is worn correctly [14]. They also conducted a comprehensive evaluation experiment on the performance of various face detectors on the face mask dataset and found that all tested models performed worse with masked faces than with unmasked faces. The density of crowds varies across different scenarios, and the types of masks worn can also vary. In many scenarios, masks worn by people are small in size and can be easily obscured. As a result, mask detection primarily involves the detection of small targets. Small target detection has many technical difficulties, such as few available features, high positioning accuracy requirements, and few datasets. There are only a few related studies on small target mask detection based on deep learning. Siradjuddin et al. improved Fast R-CNN by adding Region Proposal Network (RPN) and Region of Interest (ROI) pooling layers, and classification layers for mask detection [15]. In 2020, Bochkovskiy et al. proposed YOLOv4 [16]. Later, YOLOv5 [17] was proposed and its detection accuracy and detection speed have been improved. In [18], Moran Ju et al. proposed ARFFDet based on modified Darknet53 [19], using the scale matching strategy to select the appropriate scale and anchor size for small target detection. Zhang et al. improved Faster-RCNN and introduced considerate multi-feature for small target detection [20]. All the above studies can effectively detect faces and masks. However, for small target mask detection, most of the situations in natural scenes are complex, such as dense crowds, occluded masks, and poor ambient lighting intensity, which will reduce the detection effect of the model and fail to achieve the expected effect of crowd protection. Therefore, the mask detection model needs to have the characteristics of strong ability to detect small targets in multiple scenes.

In summary, for mask detection in different scenes, we propose a multi-scene mask detection network (YOLO-MSM) based on a multi-scale residual (MSR) block, multi-scale residual cascaded channel-spatial attention (MSR-CCSA) block, enhanced residual CCSA (ER-CCSA) block, and enhanced residual PCSA (ER-PCSA) block. Considering the performance and parameters, we use YOLOv5 as the baseline. Firstly, a convolutional neural network based on a multi-scale residual block and complementary channel-spatial attention mechanism is proposed to extract the finer features and the key information of the input image. Secondly, in order to adapt the network to different scenarios, we create a dataset containing various scenes, different crowd densities, and different types of masks. Finally, a generalization improvement strategy is introduced into the network to reduce the influence of noise on detection accuracy and improve the generalization of the model [21]. It can be seen from Figure 1 that the proposed network has the best performance.

The contributions of this paper are as follows:We propose a multi-scene mask detection network named YOLO-MSM to address the challenges posed by small target mask detection across various scenes. The network backbone incorporates the concept of the multi-scale residual to extract finer and multi-scale features. This method captures different levels of information by using multiple 3×3 convolutions in a residual block to extract features at different scales, and the equivalent receptive field increases when the input feature is fed into these 3×3 convolutions. Additionally, we incorporate cascaded and parallel channel-spatial attention blocks into the backbone and feature pyramid structure of the network. These blocks allow for the extraction of critical information from the input features while suppressing unimportant features. These methods improve the representation ability and understanding ability of the model and significantly enhance the performance of small target detection.A new dataset called Multi-Scene-Mask has been constructed, which contains various scenes, densities, and mask types. To capture the diversity of different scenarios, we construct the dataset by incorporating existing datasets such as Mask-Wearing [22], MaskedFace-Net [23], and RFMD [24], and also capture photographs of people wearing masks in multiple scenes such as hospitals, stations, campuses, parks, etc., by ourselves. The new dataset significantly improves the overall detection performance of the module in various scenarios.A model generalization improvement strategy (GIS) is presented to make the network more adaptable to images in real environments. We augment the dataset by adding white Gaussian noise to the original training set, and then retrain the network and test it on the real dataset. This approach increases the coverage, density, and difficulty of the dataset, allowing the model to learn features that are robust to small variations in the data. The experiment shows that this strategy can greatly improve the generalization of the network, and it also verifies that YOLO-MSM has stronger generalization ability.

The rest of the paper is organized as follows. In Section 2, the related research of target detection is described. Section 3 introduces the proposed method. The experiments and analysis are shown in Section 4. Finally, the conclusion and future work are described in Section 5.

## 2. Related Work

### 2.1. Object Detection

Traditional object detection tasks are based on manual feature extraction, but extracting effective features is very challenging. Furthermore, the performance of this method has been saturated. With the rapid development of deep learning in recent years, the object detection method based on deep learning gradually became mainstream [25].

Since AlexNet [26] networks emerged in 2012, convolutional neural networks (CNNs) have exploded. It has developed from the deepening of the VGG network structure to the expanding initial network [27,28,29] and Resnet [30,31] network, then to the lightweight mobile network [32] and advanced network [33]. The focus of CNN research has also shifted from parameter optimization to designing network architectures, such as using a new architecture based on the attention mechanism [34,35,36] to improve the relevant performance of networks. Deep learning-based object detection can be divided into two categories: two-stage detection and one-stage detection. The two-stage detection networks mainly include RCNN [37], SSP-Net [38], FastRCNN [39], Faster-RCNN [40], etc. Although the two-stage detection networks have a high accuracy, they still contain a series of shortcomings, including cumbersome training steps, full training speed, and occupying too much physical space during training [41]. The one-stage target detection algorithms include the YOLO series network [16,17,19,42,43,44,45], SSD network [46], and RetinaNet network [47]. Huang et al. used a generalized feature pyramid to improve the adaptability of the network [48]. Wang et al. added the SimAM space and channel attention mechanism to enhance the convergence performance of the model [49]. In [50], the detection effect is improved by adding an attention mechanism and an output layer to enhance feature extraction and feature fusion. Compared with the two-stage target detection algorithms, one-stage methods have significantly faster inference speed but may suffer from lower detection accuracy [41].

### 2.2. Small Target Detection

Thanks to the advancements in deep learning, object detection has made remarkable progress [51]. However, small target detection has remained a challenging task for a long time. Traditional object detection algorithms have demonstrated a significant gap in detecting small objects when compared to normal-sized objects. To enhance small object detection capabilities of the network, there are currently five mainstream approaches available. The first includes image enhancement represented by the scale matching strategy proposed by [52] and artificial augmentation by pasting the small objects proposed by [53]. The second includes multiple-degree learning represented by the Inside-outside Network (ION) object detection method of [54] and the extended feature pyramid network of [55]. The third includes the context learning method based on multi-scale context feature enhancement proposed by [56] and an adaptive context modeling and iterative improvement method proposed by [57]. The fourth includes the generative adversarial learning represented by GAN [58] and MTGAN [59]. The final type is the anchor-free method represented by DeNet [60] and PLN [61]. The proposal and application of these five methods greatly enhance the small target detection capability of target detection networks. However, all five aforementioned methods have significant drawbacks. Image enhancement and multiple-degree learning can generate noise during training, i.e., minor image noise may significantly impact the outcome [62]. For the multiple-degree learning method mentioned above, suitable fusion strategies and loss functions need to be designed, increasing the model’s complexity and difficulty. Context learning relies on large-scale pre-trained language models, which have high training costs and may contain some biases and errors. And it may not provide sufficient contextual information to meet the training requirements. The generative adversarial learning method requires balancing the training progress of the generator and the discriminator, avoiding the phenomenon of mode collapse or oscillation, which requires careful adjustment of hyperparameters and loss functions. When two or more target center points overlap or are close to each other, the anchor-free method may produce semantic ambiguity. Since the anchor-free method directly predicts the location and size of the target, it leads to an imbalance of positive and negative samples during training, which may affect the convergence and generalization of the model. Therefore, the five small object detection methods mentioned above have limited improvement in detection performance.

### 2.3. Mask Wearing Detection

Various studies of mask detection methods based on deep learning have been carried out in recent years. Batagelj et al. [14] constructed a large face mask dataset from the publicly available MAFA and Wider Face datasets, which made a great contribution to testing and developing mask detection models. Khandelwal et al. used the MobileNetV2 backbone to detect whether people are wearing masks [63]. Fan et al. proposed a one-stage face mask detector based on a context attention module to extract features related to the mask wearing status [64]. In [65], Qin et al. developed a new mask detection network by combining classification networks (SRCNet) and image super-resolution, and it quantified the tri-classification problem based on unconstrained 2D face images. Jiang et al. introduced landmark, a key face feature, into the detection layer of YOLOv5, improving the detection accuracy of occlusion and dense crowds [66]. In [67], a real-time CNN-based approach with transfer learning was introduced to detect if a mask was used. Asghar et al. proposed a depthwise separable convolution neural network based on MobileNet [32], which improved learning performance and decreased the number of parameters [68]. Balaji et al. employed a VGG-16 CNN model that was built with Keras/TensorFlow and Open-CV to identify the people who did not have face masks on in government workplaces [69]. In addition to the above studies, some commercial methods also provide face mask detection features. These solutions enable the easy integration of a video stream from image sensors and then apply vision techniques to monitor whether crowds are wearing masks [70,71].

## 3. Method

In order to address the issue of image noise, we weaken the feature channels that are related to noise information using a complementary attention mechanism. Furthermore, we utilize a multi-scale residual network to tackle the problem of insufficient contextual semantic information. In general, different scenes have different characteristics such as the environment, crowd density, and type of mask. Additionally, detecting small target masks remains a challenge for most scenes. To enhance the accuracy of mask detection across various scenes, we propose the YOLO-MSM network and introduce a new dataset called Multi-Scene-Mask. To improve the generalization ability of the network, we also propose a generalization improvement strategy.

### 3.1. Network Structure

#### 3.1.1. Overview of YOLO-MSM

The improved network can enhance the detection effect of the model for small objects by extracting multi-scale features and key information of input features. The overall network structure and composition diagrams of part of the module are shown in Figure 2. In this figure, the MSR block stands for the convolutional structure with multi-scale residual module. The ER-CCSA block and ER-PCSA block represent the feature extraction network with the addition of the CCSA and PCSA blocks, respectively. The MSR-CCSA block is equivalent to adding the CCSA block to the output of the MSR block.

The feature pyramid PANet has three layers of features from top to bottom. The MSR block is added into layers 2 and 4 of the backbone network without the CCSA block. The output features of layer 4 are the first layer of PANet, which are concatenated with the up-sampling features of the second layer by Concat, and the ER-PCSA block is added after convolution. Finally, the output features from the feature pyramid are fed into the detection head, and the size of the prediction box is 80×80×21. The CCSA block is added to layers 6 and 8 of the MSR block. The features output by layer 6 are used as the second layer of PANet. They concatenate with the feature detection after up-sampling in layer 3, and pass through the ER-CCSA block after convolution operation. Then, the features are concatenated with the features after the first PANet layer of down-sampling, and then input into the ER-PCSA block after convolution operation. Finally, the output features from the feature pyramid are fed into the detection head, and the prediction box size is 40×40×21. The features of layer 8 of the Res2net backbone network are input into the SPPF multi-pooling structure. The SPPF structure serializes three 5×5 max pooling layers and it can increase the receptive field and solve the target multi-scale problem in a certain length. The features output from SPPF are the third layer of PANet. Then, concatenation is performed with the features after down-sampling in the second layer of PANet. After the convolution operation, the output features are fed into the prediction head, and the prediction box size is 20×20×21. Within the above operations, the prediction boxes of three sizes are obtained, and the effective prediction of multi-scale targets is realized.

#### 3.1.2. Backbone Based on MSR and MSR-CCSA Block

YOLOv5 adopts the residual connection idea of Resnet to extract features. However, since Resnet is only a single residual operation, it is difficult to obtain subtle image features. To solve this problem, we propose a backbone network based on the fusion of the multi-scale residual (MSR) block and multi-scale residual cascaded channel-spatial attention (MSR-CCSA) block. The main advantages of this network are as follows. (1) By constructing a hierarchical residual connection in the residual block, extracting multi-scale features, increasing the receptive field of the model, and obtaining smaller features, it can improve the accuracy of the model. (2) By extracting the key information of the input features, it can suppress the interference of background information and improve the detection effect of the model for small targets. (3) By focusing on the key information of the features, it can reduce the computational complexity of the model.

To construct hierarchical residual connections in the residual blocks, inspired by Res2net, we design the MSR block. This block constructs multi-level residual connections within a residual block. It increases the convolution operation of the network. The input features are subjected to two convolution operations and then are split into *K* features. The feature i(i=2,3,⋯,K−1) is subjected to 3×3 convolution operation to extract the features of the input information, and then the extracted features are fused into the feature (i+1) in the way of residual connection. Finally, the split features after convolution are concatenated. After the above operations, we build a feature pyramid structure inside the residual block, and perform multi-scale convolution inside the feature layer. This method can take advantage of the complementarity of multi-scale features and improve the representation ability of the model. Features at different scales can capture different levels of information, for example, features at low scales can capture details and textures, and features at high scales can capture globality and semantics. There is complementarity between these features, that is, they can compensate each other for missing or insufficient information. By constructing a hierarchical residual connection in the residual block, the features of different scales can be extracted and connected in a similar way to the residual. Therefore, this method can achieve stronger multi-scale feature fusion, thereby improving the representation ability of the model. At the same time, when the input feature passes through each 3×3 convolution, the equivalent receptive field is increased, which can enhance the receptive field range of each layer of the network. And this results in many equivalent feature scales due to combinatorial effects. The larger the receptive field, the more contextual information the output feature is able to perceive. Therefore, this method can enhance the receptive field and context information of the model, and improve the understanding ability of the model. In summary, this method can improve the accuracy of the model from many aspects, mainly by exploiting the complementarity of multi-scale features, enhancing the receptive field and context information. The network structure of the MSR block is shown in Figure 3.

In real-life scenarios, images captured by cameras often contain regions that cover large areas, making object detection a challenging task. Therefore, it is crucial to focus on the extraction of key information from the image. We apply the cascaded channel-spatial attention (CCSA) module in the backbone, and it is fused with the MSR block to obtain the MSR-CCSA block. The attention mechanism can extract the key information of input features, focus on significant features, and suppress unnecessary features. In order to realize the function of joint attention of channel and space, the Convolutional Block Attention Module (CBAM) with a channel and spatial weighting scheme is used. The CBAM network structure takes the form of a series and includes a channel attention module (CAM) and a spatial attention module (SAM). In CAM, maxpooling and avgpooling operations are performed in parallel in the spatial dimension to obtain representation features. In SAM, it applies maxpooling and avgpooling, respectively, in the channel dimension to refine the channel representation information. The input feature map is passed through CAM and SAM, and multiplied with the input feature map, respectively. Finally, the new feature map is obtained. The CBAM networks can be described as follows:(1)$$F{}^{\prime }=M_{c}\left(F\right)⊗F$$
(2)$$F^{\prime \prime }=M_{s}\left(F{}^{\prime }\right)⊗F{}^{\prime }$$
where $F\in R^{C\times H\times W}$ indicates the input feature map, $M_{c}\in R^{C\times 1\times 1}$ indicates the channel attention module, $M_{s}\in R^{1\times H\times W}$ indicates the spatial attention module, ⊗ represents multiplication, and $F{}^{\prime \prime }$ indicates the output feature map. By sequentially executing the CAM and SAM, the network can learn “what” and “where” to perform in the channel and spatial dimensions, respectively. This network structure is beneficial for extracting the key information of the input features, suppressing background information, and improving the receptive field. The structure of the MSR-CCSA block can be seen in Figure 3.

#### 3.1.3. Feature Pyramid Structure Based on Complementary Attention Blocks

The feature pyramid structure repeatedly extracts three feature layers from the backbone network using an up-sampling convolutional stitching module and a down-sampling convolutional stitching module. These modules are concatenated in different cascading ways to obtain the prediction head output of three sizes and identify targets of different sizes. Up-sampling allows us to extract low-level features from the image, such as color, shape, and background texture, while down-sampling helps us extract high-level features such as faces, masks, and other complex semantics, which are more abstract. In the feature pyramid structure, we use complementary attention models, namely, the enhanced residual cascaded channel-spatial attention (ER-CCSA) block and enhanced residual parallel channel-spatial attention (ER-PCSA) block. The Bottleneck Attention Module (BAM) parallels the channel attention mechanism and spatial attention mechanism, and constructs a hierarchical attention similar to the human perception process. BAM denoises low-level features at an early stage. Then, BAM gradually focuses on the exact target, which is high-level semantics. Therefore, in order to obtain richer high-level semantics, we use BAM in the ER-PCSA block and put the block before the down-sampling operation.

The ER-CCSA block still uses CBAM to recalibrate the features. Compared with CBAM, the channel attention uses the ReLU activation function and adds a BN layer in BAM. After adaptive average pooling, multiple Linear transformations are performed to obtain the channel attention features. In spatial attention, BAM does not perform maximum and average pooling operations. Instead, it performs convolution and dilated convolution processing to increase the receptive field and make the information more abundant. Along two separate pathways, BAM can construct a hierarchical attention at bottlenecks with a number of parameters. The output feature map $F{}^{\prime }$ through BAM can be computed as follows:(3)$$F{}^{\prime }=F+F⊗\left(\sigma \left(M_{c}^{{}^{\prime }}\left(F\right)+M_{s}^{{}^{\prime }}\left(F\right)\right)\right)$$
where $F\in R^{C\times H\times W}$ indicates the input feature map, $M_{c}^{{}^{\prime }}\in R^{C\times 1\times 1}$ and $M_{s}^{{}^{\prime }}\in R^{1\times H\times W}$ represent the channel attention and the spatial attention in BAM, respectively, and σ is a sigmoid function. Both the ER-CCSA block and ER-PCSA block increase the receptive field and extract significant information on the feature map. In particular, the ER-PCSA block plays a crucial role in extracting high-level semantics. Therefore, introducing complementary attention modules into the feature pyramid can effectively improve the accuracy of the network. The network structures of the ER-CCSA block and ER-PCSA block are shown in Figure 4.

### 3.2. Construction and Synthesis of Datasets

Masks are commonly worn in various sceneries, including hospitals, stations, production workshops, and parks, among others. However, environmental characteristics, such as viewpoint, lighting, and occlusion, vary across different scenes, which can hinder the detection network’s performance. Moreover, crowd density differs across scenes, e.g., stations typically being more crowded than parks. Additionally, people wear different types of masks in different places, such as disposable surgical masks in hospitals for doctors and patients, and dust masks in production workshops for workers.

To ensure compliance with mask-wearing regulations in various scenarios, a dataset of masked faces in multiple scenarios is necessary for the training network. However, currently, there is no available dataset of multi-scene masked face images, and most of the existing masked face datasets are limited to a single scene. For these reasons, we construct a new dataset named Multi-Scene-Mask by synthesizing the existing face mask datasets Mask-Wearing, MaskedFace-Net, and RFMD. Moreover, we take photographs of mask wearing in multiple scenes and use them as the Self-Built Real Test Dataset (SBRTD) for subsequent GIS experiments. According to the environmental background, crowd density, and mask type, we divide the new dataset into three types of masked face datasets, namely, Environmental Background Masked Face Dataset (EBMFD), Crowd Density Masked Face Dataset (CDMFD), and Mask Type Masked Face Dataset (MTMFD). These three types of datasets do not include the real test set obtained by photographing. The structure of the new dataset and dataset example diagram are shown in Figure 5. The introduction of EBMFD, CDMFD, MTMFD, and SBRTD is shown below.

EBMFD: This built dataset contains 2500 masked face images, covering 8 types of environmental background. The main differences between these environments are viewpoint, lighting, occlusion, and other characteristics. The EBMFD dataset can be used to improve the detection performance of the network for masks in various backgrounds in reality.CDMFD: We divide the crowd density into four levels according to the number of face targets contained in the image. Photos containing 0 to 5 targets are used as level I, 6 to 20 as level II, 21 to 50 as level III, and 51 to 100 as level IV. The dataset contains 2000 photos of face masks, which enables the model to adapt to the detection of different crowd densities.MTMFD: In order to prevent the network from detecting only a single type of mask, we collect various types of masks and divide them into 10 categories according to shape, color, and material. The dataset has 1500 images of people wearing various types of masks.SBRTD: As we need to subsequently check the model’s generalization ability, we utilize real photos as the test dataset for the GIS experiment. We collect a total of 500 high-quality images of masked faces in various scenes, including schools, hospitals, parks, supermarkets, etc.

**Figure 5 sensors-23-08851-f005:**
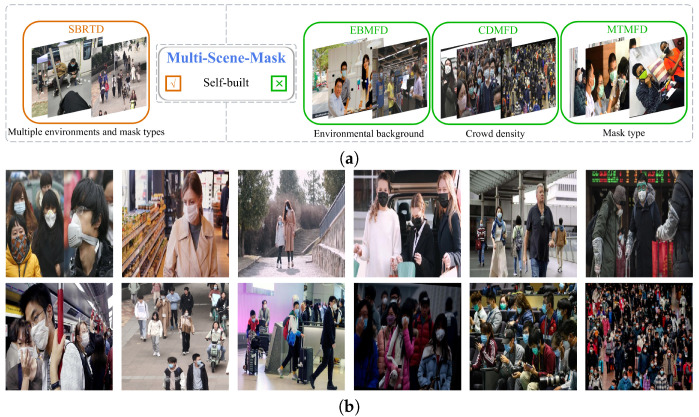
(**a**) The structure of the generated Multi-Scene-Mask dataset. (**b**) Dataset example diagram.

A total of 90% of the three types of datasets EBMFD, CDMFD, and MTMFD are used as the training set of Multi-Scene-Mask. A total of 10% of the EBMFD, CDMFD, and MTMFD datasets and the whole SBRTD dataset are used as comprehensive test sets. Among them, the SBRTD dataset is only used for the test dataset of the GIS experiment. The overall Multi-Scene-Mask statistics are presented in Table 1.

### 3.3. Generalization Improvement Strategy

For an image recognition network, when an ideal image is input, an accurate and stable recognition result can be obtained. When we add special noise to an image and feed the new image into the network, the recognition results may be wrong, even if the difference between the images before and after the noise looks slight to the human eyes. Therefore, whether it is normal noise in real life or a malicious signal against the network, we regard it as an attack on the detection network. In order to deal with these attacks, we adopt a generalization improvement strategy (GIS); during the process of training the model, we take defense into account, find loopholes, and fill them. The structure of GIS is shown in Figure 6. For the given training data,
(4)$$X=\{\left(x^{1},\hat{y^{1}}\right),\left(x^{2},\hat{y^{2}}\right),\cdots ,\left(x^{N},\hat{y^{N}}\right)\},$$
where *N* indicates that there are *N* images in this dataset, $x^{N}$ denotes the *N*th image, and $\hat{y^{N}}$ denotes the label corresponding to the *N*th image, find adversarial input $\tilde{x^{n}}$ given $x^{n}\left(n=1,2,\cdots ,N\right)$ by an attack algorithm. After going through the above process, a new training dataset is obtained, as follows:(5)$$X^{\prime }=\left\{\left(\tilde{x^{1}},\hat{y^{1}}\right),\left(\tilde{x^{2}},\hat{y^{2}}\right),\cdots ,\left(\tilde{x^{N}},\hat{y^{N}}\right)\right\}.$$

Then the new dataset is used to train the detection network. After finding the attack signal and retraining the network, the parameters of the network will change and new vulnerabilities may be created. Therefore, this process should be repeated multiple times to reduce vulnerabilities, enabling the network to effectively defend against these attack signals.

In practical applications, various factors such as image acquisition equipment and natural environment can introduce noise into real images, which can interfere with the observable information. This is different from the ideal datasets used for training the network, and obtaining high-quality images with consistent resolution can be challenging. In order to improve the generalization and robustness of the network, this paper augments the original dataset and simulates real images by adding white Gaussian noise to the original dataset. Then, we train the detection network with the augmented dataset. The structure of the augmented dataset is shown in Figure 7. White Gaussian noise is a kind of noise whose probability density function obeys a normal distribution, and can be generated by adding normally distributed random values with zero mean and standard deviation σ to the input data. Its probability density function is as follows:(6)$$f\left(x\right)=\frac{1}{\sqrt{2\pi \sigma }}e^{-\frac{{\left(x-\mu \right)}^{2}}{2\sigma^{2}}}$$
where μ is the mean, and $\sigma^{2}$ is the variance. In order to better fit the real image, the parameter selected in this paper is μ=0, σ=0.05. Adding white Gaussian noise to the original dataset is equivalent to an artificial disturbance or transformation of the data, which makes the data more complex and diverse. On the one hand, it can increase the coverage and density of the data, so that the model can be exposed to more possible situations, which effectively alleviates the problems such as data bias and data noise, thereby reducing the risk of overfitting of the model. On the other hand, adding white Gaussian noise to the input data can increase the difficulty and challenge of the data, so that the model cannot easily learn simple or superficial features from the data, and must learn deeper or essential features. This is equivalent to a method of regularization [72], which can effectively limit the complexity and redundancy of the model, thus improving the stability and interpretability of the model. In summary, by adding white Gaussian noise to the dataset, the model is forced to learn features that are robust to small variations in the input, which can help it perform better on new, unseen data. Therefore, when the processed dataset is input to the network as a model attack, the network adjusts its parameters to obtain a network with better generalization.

## 4. Experiment

### 4.1. Parameter Setting

The parameter settings of the experiments are shown in Table 2.

### 4.2. Evaluation Metrics

In order to verify the validity of the proposed model, evaluation indicators use mean average precision (mAP), mean average precision${}_{0.5}$ (mAP@0.5), mean average precision${}_{0.4}$ (mAP@0.4), mean average precision${}_{0.5:0.95}$ (mAP@0.5:0.95), detection speed (FPS), model parameters (Parameters), and model operation (GFLOPs). AP is the detection accuracy rate of a single target, which is composed of the recall rate and accuracy rate of the surrounding region. mAP is the average of all categories of AP values in the dataset, and it reflects the comprehensive performance of the model. mAP@0.5 and mAP@0.4 represent the mAP value when IoU is set to 0.5 and 0.4. mAP@0.5:0.95 represents the average mAP over IoU values from 0.5 to 0.95 with a step size of 0.05. The calculation formulas for AP, mAP, and mAP@0.5 are expressed as follows:(7)$$\mathrm{AP}=\int_{0}^{1}p\left(r\right)dr$$
(8)$$\mathrm{mAP}=\frac{\sum_{i=1}^{N}AP_{i}}{N}$$
(9)$$\mathrm{mAP@}0.5=\frac{\int_{0}^{1}p\left(r\right)dr}{N}$$
where p(r) represents the mapping relationship between precision and recall. *N* is the number of classes. FPS is the amount of image data the model can detect per second. It is used to measure the performance of the model in real time. The number of model parameters determines the size of the model file and the memory resources consumed by the model.

### 4.3. Comparison with Typical and General Detection Methods

In order to reflect the performance of YOLO-MSM, we compare our method with other typical and general detection methods, including YOLOv3, YOLOv4, the baseline network (YOLOv5), YOLOv6, YOLOv7, SSD [46], Faster R-CNN [40], and CenterNet network [73]. The experimental results are reported in Table 3.

From Table 3, it can be seen that YOLO-MSM has the highest mAP, which is 3.46% higher than the baseline. Meanwhile, the average detection speed of YOLO-MSM reached 28.16 FPS and maintained a high level. This indicates that the improved network has great real-time performance and can meet the real-time requirements of mask detection. Compared with YOLOv3, YOLOv4, YOLOv6, YOLOv7, SSD, Faster R-CNN, and CenterNet, the mAP value of YOLO-MSM is increased by 8.41%, 4.53%, 2.55%, 0.98%, 5.07%, 19.73%, and 6.16%, and has achieved the best performance in mask detection. In addition, compared with these networks, YOLO-MSM has the advantages of parameters and detection speed, as shown in Table 3. In Figure 8, the mAP@0.5 comparison curve of YOLO-MSM and the baseline is plotted. Comparing the mAP@0.5 values, it is evident that YOLO-MSM significantly enhances the average accuracy and yields a higher mAP@0.5 value than the original YOLOv5.

Figure 9 shows the confusion matrix of YOLO-MSM and the baseline network during training. The confusion matrix gives us an intuition not only about what mistakes the model is making, but also what types of errors are occurring. From Figure 9, it can be seen that YOLO-MSM has a better classification effect. The baseline is known for its difficulty in detecting certain objects, which may be incorrectly identified as background. However, this problem is effectively mitigated in YOLO-MSM. The main reason is that in most scenes, many masks are small targets, and the multi-scale residual and the attention mechanism can effectively improve the detection ability of the model for small targets. Figure 10 shows the comparison of mask detection effects of different YOLO networks in different scenes. These scenes include crowded streets, hospitals with medium density, high lighting, and special types of masks. It can be seen from Figure 10 that the proposed network in this paper has better performance for dense crowds and occluded masks in multiple complex scenes than the other YOLO networks. In general, YOLO-MSM has the best performance over other networks.

### 4.4. Comparison with Mask Detection Methods

In order to demonstrate the detection performance of YOLO-MSM more comprehensively, we compare it with existing mask detection methods. These models are PureHing [74], YOLOv5-Face [75], AIZOOTech [76], and RetinaFace [77]. Inspired by the study of Batagelj et al. [14], we test these models and YOLO-MSM on the FMLD dataset constructed by the authors. FMLD is a large face mask dataset, compiled from two publicly available datasets: MAFA and Wider Face. The dataset is partitioned into three classes: correctly worn masks, incorrectly worn masks, and without masks. It has a total of 41,934 images, of which 34,782 are used for training and 7152 for testing. The composition of FMLD is shown in Table 4. We use the test set of FMLD to test and compare the mask detection models. The comparison results are reported in Table 5.

Table 5 shows that RetinaFace performs better among the existing mask detection models. Its mAP@0.4, mAP@0.5, and mAP@0.5:0.95 values reach 88.39%, 82.02%, and 50.06%, respectively. However, its detection performance still lags behind YOLO-MSM. When the IoU is set to 0.4 and 0.5, the mAP value of YOLO-MSM reaches 91.68% and 84.92%, respectively. It also has the highest mAP@0.5:0.95 value among these models. The comparison results prove that YOLO-MSM can maintain better detection performance compared with other mask detection models in the FMLD dataset.

### 4.5. Ablation Study

To demonstrate the effectiveness of the improved network, we conduct four sets of ablation experiments on the MSR, MSR-CCSA, ER-CCSA, and ER-PCSA blocks, respectively. Each improvement scheme is described in Table 6. The network after replacing the first two Resnet modules of the backbone network with multi-scale residual modules is named YOLO-M1. After introducing the MSR-CCSA block into YOLO-M1, the network becomes YOLO-M2. Based on this, we add the ER-CCSA block into the feature pyramid and name the network YOLO-M3. YOLO-MSM is the final network model. The experimental results with an input size of 640×640 are shown in Table 7.

The results of different models for training are shown in Figure 11. As can be seen from Figure 11, each time a new block is added to the network, the average accuracy of the network is improved. It can be seen from Table 7 that after introducing the MSR block and MSR-CCSA block into the backbone network, the mAP value of the model has been improved by 0.67%. That is because after replacing Resnet with the multi-scale residual, the model can realize the complementarity of multi-scale features and enhance the receptive field and context information, so as to improve the detection accuracy of the model. On this basis, the ER-CCSA block is added in the feature pyramid to extract the key information of the features and improve the detection effect of small targets, and the mAP has been increased by 1.3%. With the ER-PCSA block in the feature pyramid, the mAP has been boosted from 96.02% to 97.51%, and the network still maintains high average detection speed at 28.16 FPS. As discussed in Section 3.1.3, BAM focuses on the exact target, namely high-level semantics, and enhances the module’s ability to refine intermediate features through two independent paths. In order to visually compare the results, a histogram based on the experimental data is plotted, as shown in Figure 12.

### 4.6. Effectiveness of GIS

In reality, due to the diversity of devices and environments, it is challenging to obtain high-quality images with consistent resolution. And most of the time, we have to face a harsh and unpredictable environment. This will have an adverse impact on the network detection accuracy. Therefore, it is necessary for us to study the model generalization of YOLO-MSM. In order to improve the generalization ability of YOLO-MSM and verify that the network has stronger generalization, this paper augments the original dataset by adding white Gaussian noise. Since we need to test the generalization of the model, we need the real test set. To this end, we use the constructed SBRTD dataset as the real test set for GIS effectiveness experiments. The augmented dataset is used to train the baseline network and YOLO-MSM, and the real dataset SBRTD is used to test the models. The test results are shown in Table 8.

As shown in Table 8, training with the original dataset, the no-mask AP value on YOLO-MSM is 95.1%, and the value for mask is 91.0%, while the AP value of the baseline in no-mask and mask detection is 91.6% and 88.5%, respectively. Compared with the baseline, the mAP value of YOLO-MSM training with the original dataset is increased by 2.0%. It can be verified that the generalization of the improved network is stronger than that of the original YOLOv5. We trained YOLO-MSM and the baseline on the augmented dataset, and then tested the networks on the SBRTD dataset. The mAP values of the baseline and YOLO-MSM are boosted by 4.3% and 2.5%, respectively. As shown in Figure 13, the AP values for no-mask and mask of the two networks are significantly improved. The above experimental results show that the generalization of the network is greatly improved after using GIS. Therefore, the model generalization improvement strategy is effective.

## 5. Conclusions and Future Work

In this paper, an improved mask detection network based on multi-scale residual and complementary attention mechanism is proposed to detect small target masks in multiple scenarios. In mask detection, different scenes have different characteristics and small target detection remains a challenge. The network backbone incorporates the concept of multi-scale residual to extract finer and multi-scale features. To this end, we propose a strategy to replace the Resnet structure of the backbone network with the MSR block. In addition, we add cascaded and parallel channel-spatial attention blocks to the backbone and feature pyramid to extract the key information of features and improve the performance of the small object detection. And experiments verify the effectiveness of the above methods in small target mask detection in multiple scenes.

For the dataset, we construct a dataset named Multi-Scene-Mask that can be used for mask detection for various environmental backgrounds, crowd densities, and mask types in multiple scenarios. It is beneficial for the development of object detection networks and image processing. Experiments are conducted on this dataset. The final mAP is 97.51%, which is much higher than other detection networks. And the FPS is 28.16. It shows that the detection speed of the network still maintains a high level after the improvement. At the same time, we establish a real dataset SBRTD to study the generalization of the network. Then, a model GIS is proposed. We augment the original dataset with white Gaussian addition noise and train the detection network with the augmented dataset. The test results verify the effectiveness of GIS and the strong generalization of YOLO-MSM.

In the future, we will focus on further performance improvement in more complex cases. The model generalization improvement strategy proposed in this paper considers only one noise case. In real life, there are other conditions such as rain, fog, low light, blurry photos, etc. We will extend the generalization improvement strategy so that the network can adapt to a more realistic real-world environment.

## Figures and Tables

**Figure 1 sensors-23-08851-f001:**
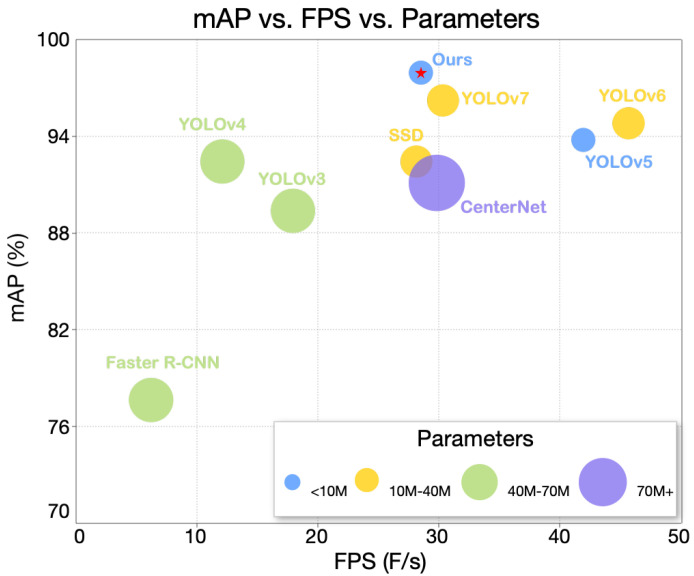
Tradeoffs of mAP, FPS, and parameters on Multi-Scene-Mask dataset. We compare the performance of our YOLO-MSM network with existing object detection networks. The results reveal that YOLO-MSM can still maintain excellent FPS and parameters while achieving the highest mAP. This is because of the combined effect of multi-scale residual and complementary attention mechanism.

**Figure 2 sensors-23-08851-f002:**
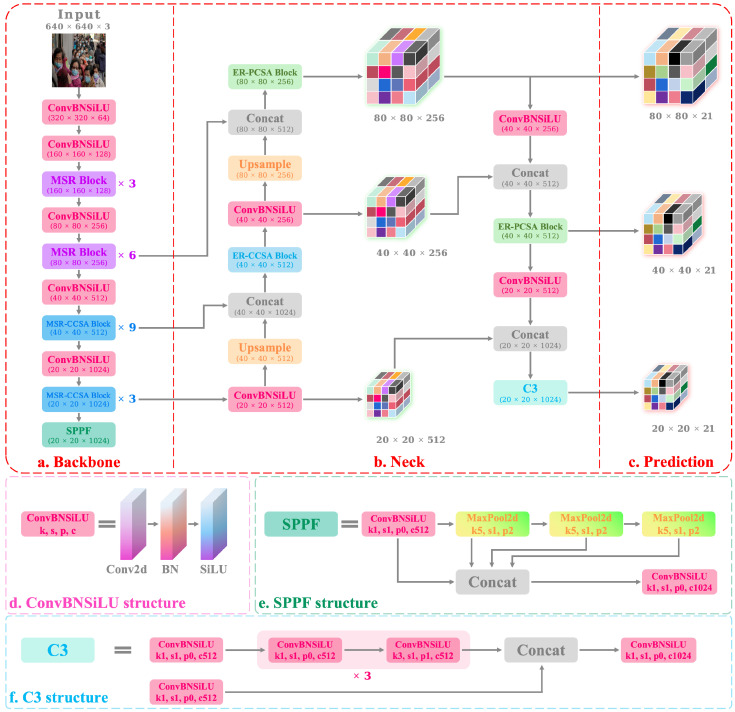
The structure of YOLO-MSM and composition diagram of part of the modules. The 320×320×64 means that the feature tensor height is 320, the width is 320, and the number of channels in the feature tensor is 64. Concat stands for concatenation. Conv2d stands for two-dimensional convolution, BN stands for batch normalization, SiLU stands for activation, MaxPool2d stands for two-dimensional max pooling. k, s, p, and c stand for kernel size, stride, padding, and number of channels, respectively. The numbers that follow indicate the values of these parameters. For example, k1 means that the size of the convolution kernel is 1×1.

**Figure 3 sensors-23-08851-f003:**
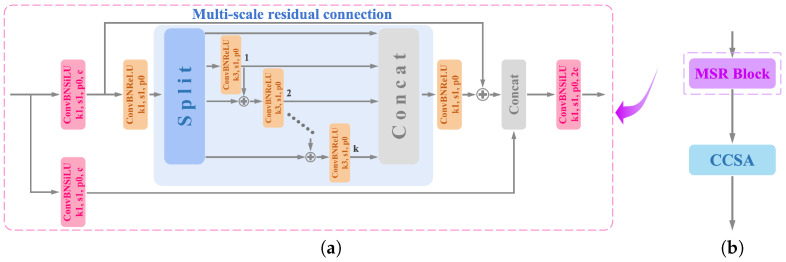
(**a**) The structure of MSR block. ReLU in ConvBNReLU represents the activation function. Split represents the operation of splitting the feature tensor into *K* pieces. Concat stands for concatenation. (**b**) The structure of MSR-CCSA block. CCSA can obtain the refined features by performing a product between the input features and the obtained attention maps.

**Figure 4 sensors-23-08851-f004:**
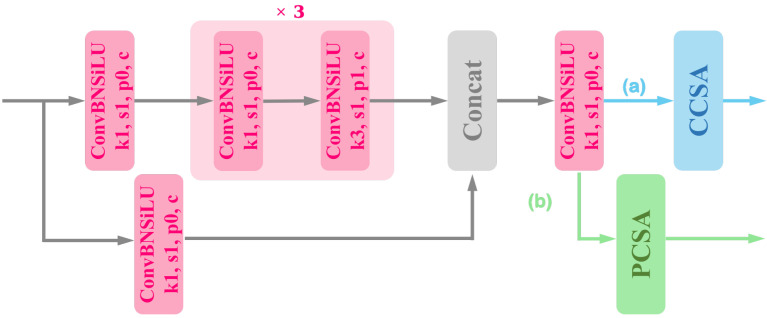
The structure of ER-CCSA block and ER-PCSA block. The structure diagram consists of two independent paths (**a**,**b**). When path (**a**) is selected, the structure is represented as ER-CCSA block. When path (**b**) is selected, this structure is represented as ER-PCSA block.

**Figure 6 sensors-23-08851-f006:**
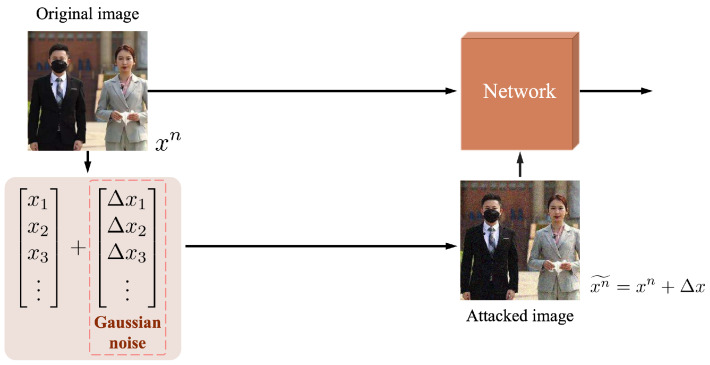
The structure of GIS. $x^{n}$ denotes the *n*th image in the original dataset. $x_{1},x_{2},x_{3},\cdots$ denote the features of this image. $\Delta x_{1},\Delta x_{2},\Delta x_{3},\cdots$ denote the white Gaussian noise added to the corresponding feature. $\tilde{x^{n}}$ stands for the image after adding noise. The original dataset and the new dataset processed with Gaussian noise jointly train the detection network to improve the generalization.

**Figure 7 sensors-23-08851-f007:**
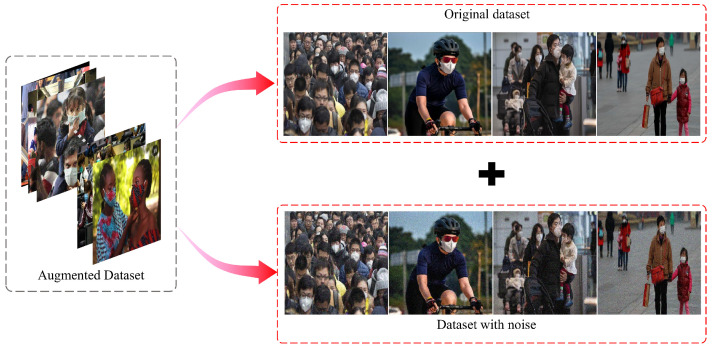
The structure of the augmented dataset. The dataset consists of the original dataset and the noise-added dataset.

**Figure 8 sensors-23-08851-f008:**
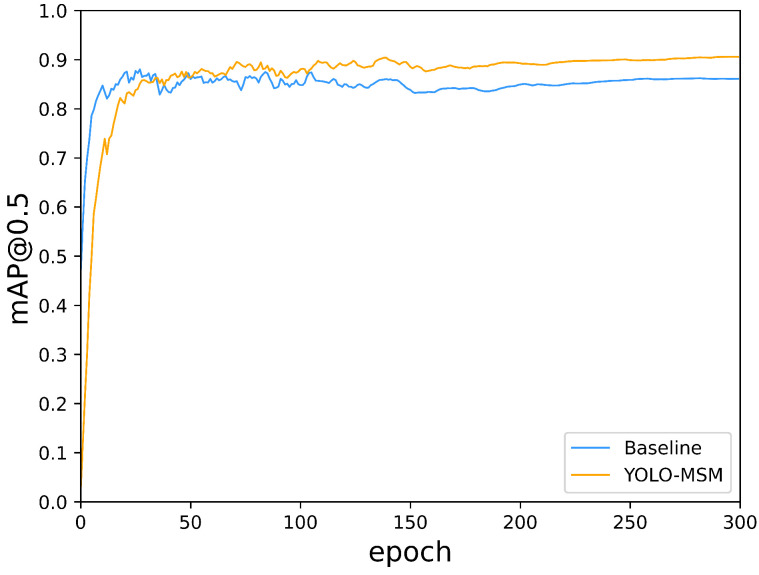
mAP@0.5 construct curve of YOLO-MSM and the baseline. The horizontal axis represents the epoch, while the vertical axis represents the values of mAP@0.5.

**Figure 9 sensors-23-08851-f009:**
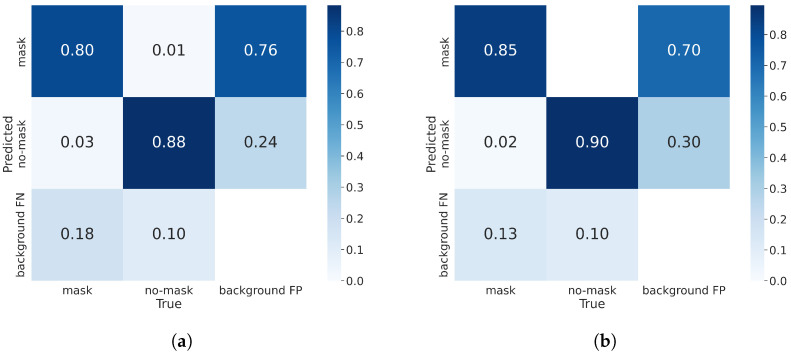
The comparison of the confusion matrix between the baseline network and YOLO-MSM. (**a**) The confusion matrix of the baseline. (**b**) The confusion matrix of YOLO-MSM.

**Figure 10 sensors-23-08851-f010:**
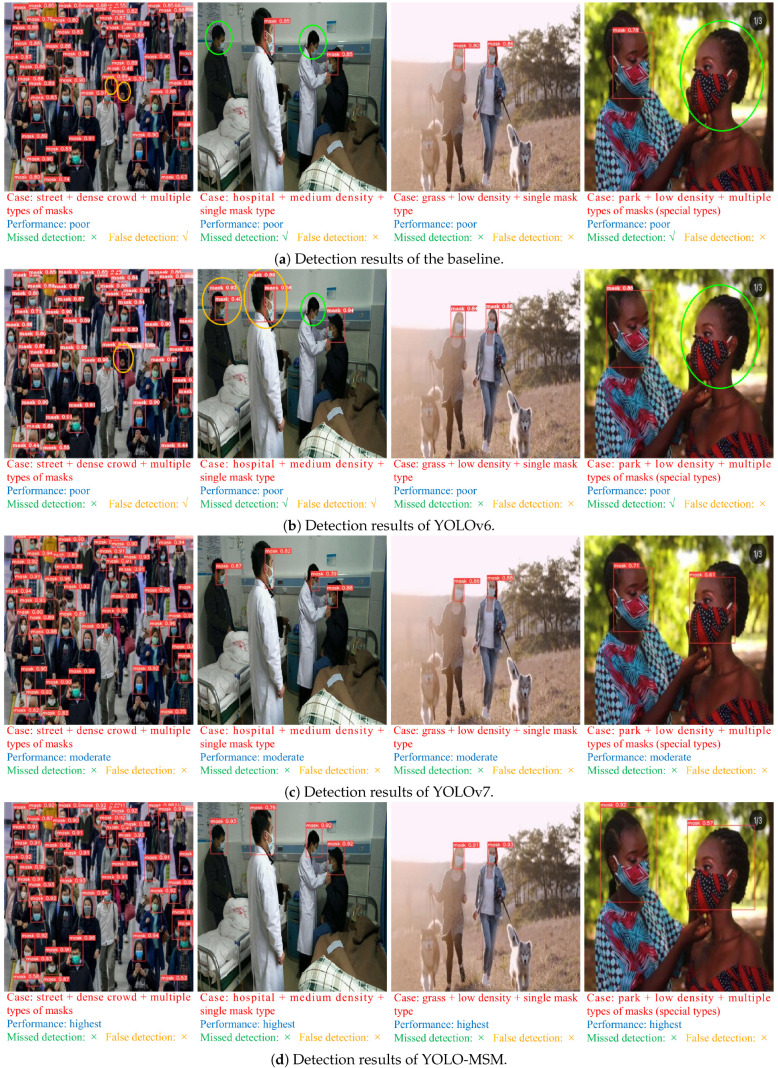
Comparison of mask detection results for different backgrounds, crowd densities, and mask types.

**Figure 11 sensors-23-08851-f011:**
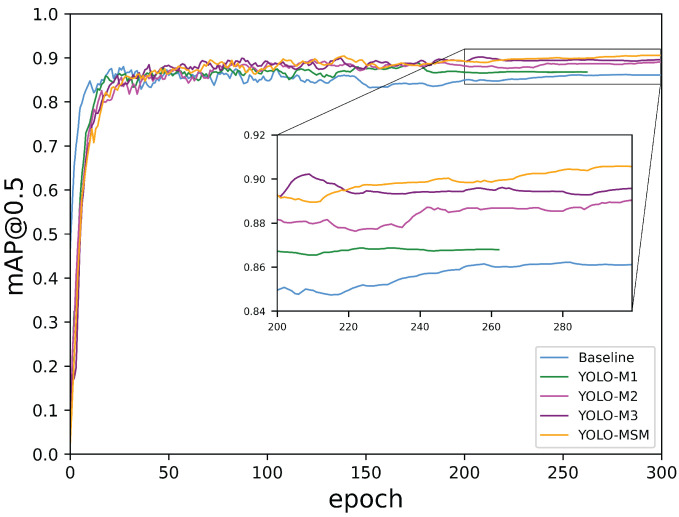
mAP@0.5 curves of different models. The rectangular area shows the local enlarged drawing for training epochs from 200 to 300.

**Figure 12 sensors-23-08851-f012:**
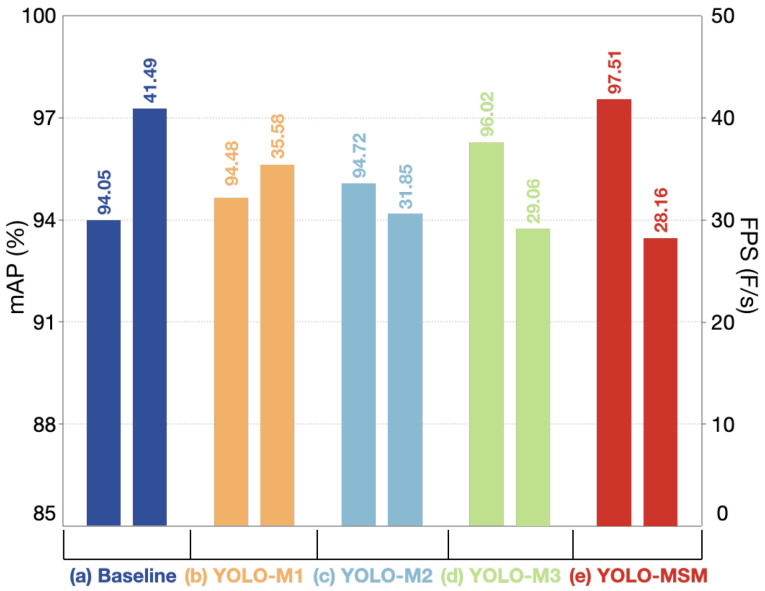
Histogram of ablation experiment results. The horizontal axis represents the types of models. The vertical axis on the left represents the values of mAP and the vertical axis on the right represents the values of FPS.

**Figure 13 sensors-23-08851-f013:**
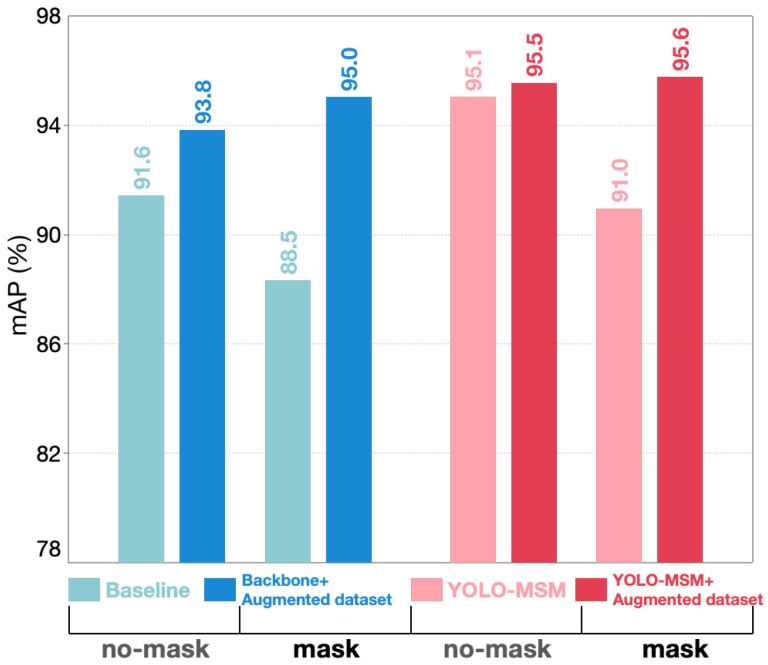
Histogram of GIS effectiveness results. The experimental results are obtained by testing the network on the SBRTD dataset.

**Table 1 sensors-23-08851-t001:** Overall Multi-Scene-Mask statistics.

Dataset	Purpose	Images	Faces	Labels	
				With Mask	Without Mask
EBMFD	Train	2250	5769	5013	756
	Test	250	638	561	77
CDMFD	Train	1800	30,469	28,796	1673
	Test	200	3537	3120	417
MTMFD	Train	1350	2268	2219	49
	Test	150	224	207	17
SBRTD	Train	0	0	0	0
	Test	500	2183	1864	319
Muti-Scene-Mask	Train	5400	38,506	36,028	2478
	Test	1100	6582	5752	830
	Totals	6500	45,088	41,780	3308

**Table 2 sensors-23-08851-t002:** Parameters of the training process.

Training Config	Settings
Dataset	Multi-Scene-Mask
GPU	NVIDIA GeForce RTX3080 (Nvidia Corporation, Santa Clara, CA, USA)
Deep learning framework	Pytorch 1.13.0
Programming language	Python 3.9
Image size	640×640
Batch size	16
Epoch	300
Optimizer	SGD
Base learning rate	0.01
Learning rate momentum	0.937
Weight decay	$5\times 10^{-4}$

**Table 3 sensors-23-08851-t003:** Performance comparison results of different methods.

Methods	mAP (%)	Parameters/10${}^{6}$	FPS (F/s)	Inference Time (s)
Faster R-CNN	77.78	60.61	6.04	0.1655
CenterNet	91.35	86.67	28.55	0.0350
SSD	92.44	26.79	27.67	0.0361
YOLOv3	89.10	61.50	17.36	0.0576
YOLOv4	92.98	64.01	11.32	0.0883
YOLOv5	94.05	7.02	41.49	0.0241
YOLOv6	94.96	18.50	45.30	0.0221
YOLOv7	96.53	36.49	30.03	0.0333
**YOLO-MSM (Ours)**	**97.51**	**7.14**	**28.16**	**0.0355**

**Table 4 sensors-23-08851-t004:** The composition of FMLD. The faces in the detaset are partitioned into three classes: correctly worn masks, incorrectly worn masks and without masks.

Dataset	Purpose	Images	Faces	Labels		
				Correctly Worn Mask	Incorrectly Worn Mask	Without Mask
MAFA	Train	25,876	29,452	24,603	1204	3645
	Test	4935	7342	4929	324	2089
Wider Face	Train	8906	20,932	0	0	20,932
	Test	2217	5346	0	0	5346
FMLD	Train	34,782	50,384	24,603	1204	24,577
	Test	7152	12,688	4929	324	7435
	Totals	41,934	63,072	29,532	1528	32,012

**Table 5 sensors-23-08851-t005:** Results of comparison with different mask detection methods.

Methods	mAP@0.4 (%)	mAP@0.5 (%)	mAP@0.5:0.95 (%)
PureHing	75.38	70.12	35.48
YOLOv5-Face	77.21	74.58	41.65
AIZOOTech	85.47	78.29	47.75
RetinaFace	88.39	82.02	50.06
**YOLO-MSM (Ours)**	**91.68**	**84.92**	**51.42**

**Table 6 sensors-23-08851-t006:** Different improvement schemes. “✓” indicates the selected module.

Methods	MSR Block	MSR-CCSA Block	ER-CCSA Block	ER-PCSA Block
YOLO-M1	✓			
YOLO-M2	✓	✓		
YOLO-M3	✓	✓	✓	
**YOLO-MSM (Ours)**	**✓**	**✓**	**✓**	**✓**

**Table 7 sensors-23-08851-t007:** Results of ablation experiments.

Methods	mAP (%)	mAP@0.5 (%)	Parameters/10${}^{6}$	FPS (F/s)
Baseline	94.05	86.17	7.016	41.49
YOLO-M1	94.48	86.79	7.032	35.58
YOLO-M2	94.72	88.91	7.090	31.85
YOLO-M3	96.02	89.57	7.131	29.06
**YOLO-MSM (Ours)**	**97.51**	**90.56**	**7.141**	**28.16**

**Table 8 sensors-23-08851-t008:** Test results of GIS effectiveness experiments. “✓” indicates that the network is trained on the augmented dataset; “×” indicates that the network is trained on the original dataset.

Methods	Augmented Dataset	No-Mask (%)	Mask (%)	mAP (%)	mAP@0.5 (%)
Baseline	×	91.6	88.5	90.1	87.3
	✓	93.8	95.0	94.4	89.8
**YOLO-MSM (Ours)**	×	95.1	91.0	93.1	90.3
	**✓**	**95.5**	**95.6**	**95.6**	**90.9**

## Data Availability

The data presented in this study are available on request from the corresponding author.

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
