# Peer review of "Multi-Scene Mask Detection Based on Multi-Scale Residual and Complementary Attention Mechanism"

_sensors, 2023, doi:10.3390/s23218851_

Round 1
Reviewer 1 Report
Comments and Suggestions for Authors
In this manuscript, the authors propose the YOLO-MSM network tailored for a multi-scene mask detection task. This network is anchored around multi-scale residual (MSR) Block, multi-scale residual cascaded channel-spatial attention (MSR-CCSA) Block, enhanced residual CCSA (ER-CCSA) Block, and the enhanced residual PCSA (ER-PCSA) Block. The experimental results provided compelling evidence supporting the effectiveness of the presented method. My minor concerns are as follows:
1. For some sentences in the manuscript, the corresponding references should be added. E.g., “Deep learning for mask detection has important demand in medical and industrial production, which reflects the application of neural networks and image sensors in daily life.”, “Although the two-stage detection networks...too much physical space during training.”, etc.
2. Some phrasing can benefit from revisions for clarity and completeness. Specifically, within sub-section 2.2, the statement: “However, all five aforementioned methods have significant drawbacks. Image enhancement and multiple-degree learning can generate noise during training [56] and have high computational costs.” The shortcomings of these five methods for small target detection should be added.
3. There are areas where the figures might lead to misinterpretations. For instance, figure3(a) should show that the input features are split into k features, but the figure may make people mistakenly think that the input features are constrained to be split into 3 features.
4. The enhancement in model accuracy, attributed to the use of hierarchical residual connections within the residual blocks, requires a more comprehensive explanation. In addition, it also should be mentioned in the introduction.
5. The rationale behind the assertion that the generalization improvement strategy bolsters the network's generalization capabilities should be explained in more detail.
6. In the experimentation section, there is a reference gap. “...the baseline network (YOLOv5), YOLOv6, YOLOv7, SSD [40], Faster R-CNN [35] and CenterNet network”. The citation for CenterNet should be added.
7. More references published in recent three years can be added.
Reviewer 2 Report
Comments and Suggestions for Authors
Overall well written article.
No corrections needed.
Reviewer 3 Report
Comments and Suggestions for Authors
Reword the sentence so that it does not start with a reference:
[10] improved Fast R-CNN by adding Region Proposal Network (RPN) and Region of Interest (ROI) pooling layers, and classification layers for mask detection.
It is better to name the authors of the paper rather than just having a reference.
In 2020, [11] proposed YOLOv4.
In the introduction, I miss the mention of methods that try to determine whether a person is wearing a correctly worn mask, in addition to mask detection. The main reference in this area is the paper How to Correctly Detect Face-Masks for COVID-19 from Visual Information? The authors have also published a method and an annotated FMLD (The Face Mask Label Dataset) database: https://github.com/borutb-fri/FMLD
The right comparison of your method would be not with general object detection methods but with methods that detect masks.
When creating or testing a database, consider also the FMLD database with 31,060 faces with masks.
In related work, a more in-depth investigation of all methods working in the field of mask detection could also be done.
A table with numbers of how many images/faces are in what category in the database would also be useful.
Round 2
Reviewer 3 Report
Comments and Suggestions for Authors
There is an error in the cover letter in Table 1 in the naming of the database. It is corrected in the article.